# Quantitative Metabolomic Analysis of Changes in the Rat Blood Serum during Autophagy Modulation: A Focus on Accelerated Senescence

**DOI:** 10.3390/ijms232112720

**Published:** 2022-10-22

**Authors:** Olga Snytnikova, Yuri Tsentalovich, Renad Sagdeev, Nataliya Kolosova, Oyuna Kozhevnikova

**Affiliations:** 1International Tomography Center, Siberian Branch of the Russian Academy of Sciences, Institutskaya Str. 3a, 630090 Novosibirsk, Russia; 2The Federal Research Center Institute of Cytology and Genetics, Siberian Branch of the Russian Academy of Sciences, Academician Lavrentiev Avenue, 10, 630090 Novosibirsk, Russia

**Keywords:** autophagy, blood serum, senescence-accelerated OXYS rats, chloroquine, metabolomics, NMR spectroscopy

## Abstract

Autophagy is involved in the maintenance of cellular homeostasis and the removal of damaged proteins and organelles and is necessary to maintain cell metabolism in conditions of energy and nutrient deficiency. A decrease in autophagic activity plays an important role in age-related diseases. However, the metabolic response to autophagy modulation remains poorly understood. Here, we for the first time explored the effects of (1) autophagy activation by 48 h fasting, (2) inhibition by chloroquine (CQ) treatment, and (3) combined effects of fasting and CQ on the quantitative composition of metabolites in the blood serum of senescent-accelerated OXYS and control Wistar rats at the age of 4 months. By means of high-resolution ^1^H NMR spectroscopy, we identified the quantitative content of 55 serum metabolites, including amino acids, organic acids, antioxidants, osmolytes, glycosides, purine, and pyrimidine derivatives. Groups of 48 h fasting (induction of autophagy), CQ treatment (inhibition of autophagy), and combined effects (CQ + fasting) are clearly separated from control groups by principal component analysis. Fasting for 48 h led to significant changes in the serum metabolomic profile, primarily affecting metabolic pathways related to fatty acid metabolism, and led to metabolism of several amino acids. Under CQ treatment, the most affected metabolites were citrate, betaine, cytidine, proline, tryptophan, glutamate, and mannose. As shown by two-way ANOVA, for many metabolites the effects of autophagy modulation depend on the animal genotype, indicating a dysregulation of metabolome reactivity in OXYS rats. Thus, the metabolic responses to modulation of autophagy in OXYS rats and Wistar rats are different. Altered metabolites in OXYS rats may serve as potential biomarkers of the manifestation of the signs of accelerated aging. Metabolic signatures characteristic to fasting and CQ treatment revealed in this work might provide a better understanding of the connections between metabolism and autophagy.

## 1. Introduction

Autophagy is a process of intracellular self-destruction that balances synthesis and degradation. During autophagy, the cytoplasmic material is delivered into lysosomes and then unnecessary or damaged cellular components are disposed of in order to maintain cellular function [1]. This allows cells to adapt to stress, mobilize their energy reserves, and decompose potentially harmful components.

Studies show that autophagy is involved in various processes from fighting bacterial and viral infections to cell renewal in a developing embryo; autophagy is also one of the main mechanisms for maintaining cellular and organismal homeostasis under conditions of starvation, diabetes, cardiovascular and infectious diseases, neurodegenerative diseases, disorders, and neoplasms [2,3,4,5,6,7,8,9,10,11]. Numerous studies have found that autophagy plays an important role in the pathogenesis of many age-related diseases.

The study of the functional and regulatory aspects of autophagy has attracted increasing interest in recent years [4,12,13]. In recent reviews, the processes of molecular regulation of homeostatic autophagy are considered in detail, and the main inducers and inhibitors of autophagy are presented, indicating the mechanisms of action [14,15,16,17]. Autophagy is quickly upregulated in response to a variety of stressors, such as starvation, organelle or DNA damage, hypoxia, endoplasmic reticulum stress, or infection [18]. Most often, researchers use fasting to induce autophagy, which is a strong physiological stimulus for macroautophagy/autophagy [19,20]. Fasting induces extensive metabolic adaptation at the cellular level and leads to the induction of autophagy throughout the body. Autophagic degradation ensures the recycling of resources and contributes to the maintenance of the energy homeostasis. It has been shown that fasting for 2–4 days leads to dramatic changes in the blood metabolome in both humans and experimental animals. There is a significant increase in the amount of free fatty acids, acylcarnitines, cystine, and some amino acids, as well as a significant decrease in the levels of tryptophan, choline phosphate, hippuric acid, and glycerophosphocholine [19,20,21,22,23]. Autophagy suppression may result in the accumulation of cellular and extracellular waste such as lipofuscin, amyloid, and atherosclerotic plaques. Apparently, such processes lead to metabolic disturbances which can be determined using metabolomics. Metabolites participate in the regulation of autophagy and could be biomarkers and/or targets for autophagy-related processes [15]. The use of modern omics technologies, such as metabolomic profiling, makes it possible to obtain information about metabolic changes in the body during autophagy, to make a significant contribution to the deciphering of the complex of regulatory mechanisms involved in autophagy, and, in addition, to contribute to the understanding of mechanisms responsible for the development of autophagy-related diseases [15]. Therefore, the metabolomic profiling might be the main tool for autophagy research. However, to date, the results of metabolomic studies are mostly present as a qualitative or semi-quantitative comparison of metabolomic data [19,20,23,24]. 

One way to study autophagy is its modulation through induction by starvation and inhibition by autophagy inhibitors. One of the known inhibitors of autophagy is chloroquine (CQ), a compound from the group of 4-aminoquinoline derivatives, which blocks the stage of fusion of autophagosomes with lysosomes [25]. Due to its immunosuppressive and anti-inflammatory effects, CQ is used as a medication for autoimmune-related diseases, such as rheumatoid arthritis and systemic lupus erythematosus, as well as a sensitizing agent for specific cancers [26]. At present, the effects of CQ administration on the metabolome are not well understood.

In this work, we used senescence-accelerated OXYS rats and control Wistar rats. The OXYS rat strain is a unique genetic model of accelerated aging established by selection and inbreeding of Wistar rats sensitive to the cataractogenic effects of galactose [27]. From a young age (3–4 months), the OXYS rats spontaneously develop early signs of age-related pathologies, which progress with age and reproduce the features typical for human diseases, such as cataract, retinopathy, osteoporosis, arterial hypertension, accelerated thymus involution, sarcopenia, and brain neurodegenerative changes [28,29,30,31]. Previously, the changes in the activity of the mTOR signaling pathway were detected in OXYS rats [31], which may indicate a violation of autophagy. It has recently been shown that the onset and progression of retinopathy in OXYS rats occurs against the background of changes in the expression of genes associated with the autophagy process and a reduction in autophagic flux in response to stress [29]. In the past, an exhaustion of autophagy processes was detected in the kidneys of OXYS rats [32], consistent with their accelerated-senescence phenotype. A reduced ability to increase autophagy activity in response to metabolic stress may be an important sign in the pathogenesis of age-related diseases [29].

The main goal of the present work was to study the quantitative changes in the concentrations of metabolites under conditions of induction and inhibition of autophagy in blood serum of senescent-accelerated OXYS and Wistar at the age of 4 months. The method of high-field NMR spectroscopy was used to study changes in the serum metabolomic composition in the following experimental groups (Figure 1): (1) intact control, (2) induction of autophagy by 48 h fasting, (3) phosphate-buffered saline (PBS) treatment as a vehicle control, (4) inhibition of autophagy by intraperitoneal (i/p) injections of CQ, (5) combined effects of fasting and CQ treatment. Here, for the first time we report the alterations of level of metabolites in serum of senescent-accelerated OXYS rats and normal Wistar rats during autophagy modulation. 

## 2. Results

### 2.1. Quantitative Metabolomic Profiling of Blood Serum

Quantitative metabolomic profiling of rat blood serum was carried out by ^1^H NMR spectroscopy. A typical NMR spectrum of the protein-free serum extract is shown in Appendix A. In each sample, 55 metabolites were identified, including amino acids, organic acids, antioxidants, osmolytes, glycosides, purine, and pyrimidine derivatives. Appendix A presents a list of identified and quantified metabolites with their chemical shifts and multiplicities. The determined concentrations of metabolites are in the range from 1 µM to 10 mM. The range and the mean values of metabolite concentration (µM) in the rat blood serum of groups employed in autophagy experiments are collected in Appendix A. 

To evaluate the differences in the metabolomic profiles of studied groups, we performed principal component analysis (PCA) using normalized (autoscaled) concentration data. Figure 2 shows the PCA score plot for concentration of aqueous metabolites extracted from blood serum of Wistar and OXYS rat strains, according to experimental design (Appendix A).

Figure 2 shows that fasting induces significant metabolomic changes along the first principal component (PC1), while changes related to CQ administration mostly correspond to PC2. CQ influence is noticeable for both fasting animals and for the group of animals without food restriction. The intact control and vehicle (animals treated with PBS injection) groups have a significant overlap in the distribution areas, indicating that the metabolomic profile of the blood serum does not change when exposed to PBS injections. For a better visualization of a difference between groups, Appendix A shows the results of sparse partial least squares discriminant analysis (sPLS-DA) of the same data. 

### 2.2. Changes in Metabolomic Profile during Fasting

A comparison of metabolomic profiles of blood serum of groups of animals without dietary restrictions and fasting for 48 h for both lines of rats is shown in Figure 3. PCA score plots show a significant difference between the metabolomic profiles of fasting and control groups. The distribution areas are well separated for both Wistar and OXYS rats. The comparison between fasting and control groups was performed using one-way ANOVA. We observed a statistically significant increase under fasting for 12 compounds in Wistar rats and 11 compounds in OXYS; the decrease was found for 12 metabolites in Wistar and 4 metabolites in OXYS (Appendix A). Volcano plots (Figure 3B,D) show the metabolites with the highest (fold change >1.5) and statistically significant (*p* < 0.05) differences between fasting and control groups. The boxplots for some metabolites are presented in Figure 4, and the remaining boxplots are given in Appendix A.

The most dramatic changes in the serum metabolomic composition correspond to the increase in the concentrations of ketone bodies and organic acids: acetone, acetoacetate, ketoleucine, alpha-aminobutyric acid (AABA), 2-hydroxybutyrate, 3-hydroxybutyrate, isobutyrate. In particular, the elevation of the 3-hydroxybutyrate level was more than 20-fold from 80 µM in control animals to approximately 2 mM in rats with the dietary restrictions. This may indicate the state of ketosis in rats after 48 h of fasting. The most significant decrease (more than two-fold) we observed for proline, carnitine, cytidine, and uridine. The set of differential metabolites for Wistar and OXYS rats are similar, but two-way ANOVA indicates that for several metabolites the effect of fasting depends on genotype. Serum concentrations of glycine, acetylcarnitine, and α-aminobutyrate after fasting increased in rats of both strains, but their increase was higher in OXYS rats as compared to Wistar rats. The levels of tyrosine and glucose significantly decreased, and that of 2-ketoisovalerate increased during fasting in Wistar rats, but not in OXYS rats. The concentration of ketoleucine after starvation increased in rats of both strains, but in Wistar rats the increase was more significant. In Wistar rats, the level of alanine after fasting was reduced more significantly than in OXYS rats. In control OXYS rats, the levels of glutamate and lactate were significantly lower than in Wistar rats and did not change after fasting, in contrast to Wistar rats, in which their levels significantly decreased (Figure 4 and Appendix A).

The Metabolite Set Enrichment Analysis (MSEA, Figure 5) shows that for both rat strains, the most affected metabolic pathways are fatty acid metabolism, beta oxidation of very long chain fatty acids, and metabolism of glycine, serine, glutamate, alanine, arginine, proline, and glutathione.

### 2.3. Changes in Metabolomic Profile under Chloroquine Treatment

To elucidate the effect of chloroquine on the metabolomic profile of rat blood serum, we compared groups of control animals and animals injected with chloroquine. Figure 6 (score plots A,C) shows a good separation between control and CQ-injected rats. According to the univariate analysis, CQ administration increased the concentrations of six metabolites in the blood serum of rats (cytidine, mannose, 2-hydroxy-3-methylbutyrate, phenylalanine, creatine). Exposure to chloroquine reduced the concentration of thirteen metabolites; for four compounds the decrease was more than two-fold (citrate, betaine, proline). Two-way analysis of variance shows that for many metabolites, the effect of chloroquine depends on the genotype of animals (cytidine, β-mannose, tryptophan, sarcosine, methionine, glucose, betaine, asparagine). The metabolites with the highest difference between control and CQ groups are highlighted in Volcano plots (Figure 6B,D).

According to MSEA (Figure 5), the effects of CQ administration for both Wistar and OXYS strains are similar; the most affected metabolic pathways are the citric acid cycle and metabolism of arginine, proline, glycine, serine, methionine, betaine, cysteine, folate, and tyrosine. 

The effect of CQ on the serum metabolome of fasting rats is similar to that of control animals (Figure 7) and leads to significant metabolomic changes—an increase in citrate, betaine, and pyruvate and a decrease in mannose and 2-hydroxy-3-methylbutyrate, in particular.

### 2.4. Changes in Metabolomic Profile Depending on the Genotype

Pairwise comparison of the metabolomic profiles of the following groups was carried out using PCA analysis: control groups, groups with autophagy induction by fasting, groups with autophagy inhibition by CQ, and combined effects of fasting and CQ depending on the genotype of animals (Appendix A). According to PCA analysis, the serum metabolomes of Wistar and OXYS rats after exposure to CQ were similar: there was a large overlap in the distribution areas of the samples. For groups of control, fasting, and combined exposure, there was a partial overlap of distribution areas, indicating that the basic metabolomic profiles of Wistar and OXYS rats and their reactivity in response to fasting differ.

We performed a two-way analysis of variance (two-way ANOVA), which showed that changes in the concentration of 20 metabolites in the autophagy modulation experiment significantly depend on the genotype of animals (see Figure 4 and Appendix A). In particular, independent of the experimental stage, serum of prematurely aging OXYS rats as compared to Wistar contained elevated levels of side-branched-chain amino acids (BCAAs) isoleucine, leucine, and valine, and also of tyrosine, sarcosine, and histidine, and reduced concentrations of cytidine and 3-methyl-2-oxovalerate. OXYS rats had a reduced response to CQ inhibition for α- and β-mannose; decreased fasting response for 2-ketoisovalerate, ketoleucine, and glucose; and an increased response to fasting for α-aminobutyrate, glycine, isoleucine, acetylcarnitine, and serine. At the same time, intact Wistar and OXYS rats (control groups) significantly differed in the content of five metabolites in the blood serum: glutamate, lactate, citrate, cytidine, and sarcosine.

## 3. Discussion

The aim of this study was to investigate the changes in the metabolomic composition of blood serum during the induction and inhibition of the autophagy process. We used two strains of Wistar and OXYS rats, the latter being a unique genetic model of accelerated senescence and the development of age-related diseases [27]. This allowed us to demonstrate changes in the blood serum metabolome not only under the condition of autophagy processes, but also in pathology. To our knowledge, this work is the first report on the serum metabolome of fasting rats studied by ^1^H NMR spectroscopy. We assessed the alterations of quantitative content of 55 serum metabolites, including amino acids, organic acids, antioxidants, osmolytes, glycosides, purine, and pyrimidine derivatives. Below we discuss the obtained results of effects of autophagy modulation by fasting and CQ treatment on the serum metabolome, as well as differences in the metabolic response to modulation of autophagy between the senescence-accelerated OXYS rat strain and Wistar rats with normal aging rate.

### 3.1. Effects of Fasting

Metabolomic analysis of rat serum performed in this work demonstrates that 48 h fasting leads to significant changes in the serum metabolomic profile. In particular, we observed the elevation of levels of ketone bodies and butyric acids, indicating the development of ketosis in experimental animals. The strongest growth (approximately 20-fold) was found for 3-hydroxybutyrate. The observed metabolomic changes indicate that starving mostly affects the metabolic pathways related to fatty acid metabolism and to metabolism of several amino acids (Figure 5). Our findings are in agreement with previous studies reporting serum metabolic changes after starvation [19,20,33,34,35]. Pietrocola et al. showed that starvation of humans for several days caused a dramatic increase in multiple distinct free fatty acids, acylcarnitine species, and oxidized amino acid dimer cystine, as well as a significant decrease in tryptophan, choline phosphate, hippuric acid, and glycerophosphocholine in plasma [19]. In mice, metabolomic changes induced by a 48 h starvation period in the plasma included an increase in concentration of multiple acylcarnitine species (including hydroxybutyrylcarnitine), free fatty acids, and glucocorticoids, and a significant decrease in the concentrations of several amino acids (arginine, methionine, proline, etc.) [19]. Ketone bodies (acetoacetate and β-hydroxybutyric acid) were highly accumulated in 48 h fasting mice. Serum levels of many amino acids, glutathione, ornithine, and citrulline, were decreased, and cysteine, homocysteine, and taurine were elevated; ketone bodies (acetoacetate and β-hydroxybutyric acid) were highly accumulated in 48 h fasting mice [20]. The comparison of the metabolomic profiles between the fed and fasted state in pigs revealed differences in 15 compounds, most of which were not significantly different between 24 h fasting and 48 h fasting, and which are involved in linoleic acid metabolism, amino acid metabolism, sphingolipid metabolism, and pantothenate and CoA biosynthesis [35]. Interestingly, metabolomic changes in serum induced by overnight 16 h fasting in rats were modest in magnitude but broad in extent, with up to one-half of monitored metabolites significantly affected [34]. Our data are in a good agreement with the study of metabolomics of prolonged fasting in humans [36]. In this study, new metabolites were identified as fasting markers, such as 2-hydroxybutyrate and α-aminobutyrate, as well as methionine and the branched chain ketoacids that all relate to perturbations in amino acid catabolism [36]. We also found significant increase in α-aminobutyrate, 2-hydroxybutyrate, and branched-chain ketoacids during 48 h fasting in serum of rats. Around 44% of metabolites were shown to change significantly in response to prolonged fasting from 12 to 36 h [36]. Notably, the number of metabolites significantly affected by fasting are similar to or greater than the number affected by toxicants inducing profound histopathological changes [34].

### 3.2. Effects of Autophagy Inhibition by CQ

At the present, the metabolic response to autophagy inhibition is not well understood. In this work, we inhibited the autophagy with CQ injections. CQ is a widely used inhibitor of autophagy both in cell culture and in vivo by targeting lysosomes. A recent study showed that CQ inhibits autophagy by disrupting the fusion of autophagosome with lysosomes [25]. Chloroquine has been used to treat systemic lupus erythematosus and rheumatoid arthritis, but long-term treatment is known to lead to retinal toxicity [37,38]. Early reports suggested some benefit for COVID-19, although this hypothesis has not been confirmed [39]. Several clinical trials have shown that CQ and its derivate hydroxychloroquine enhance the potential of combinatorial anti-cancer therapy by enhancing the sensitivity of tumor cells, although its exact mechanisms of action remain unclear [25]. Since CQ is currently considered as a sensitizing agent for specific cancers and a key compound used in clinical trials aimed to treat tumors by autophagy inhibition [26], knowledge of its effects on the metabolome under various conditions is relevant and important. Previously, it has been shown that inhibition of autophagy by CQ leads to significant alterations in Krebs cycle intermediates, especially those downstream of citrate synthase and associated with glutaminolysis, and results in the decrease in citrate synthase activity [26]. Our results are in a good agreement with this study: we also showed a significant (almost two-fold) reduction in serum citrate and glutamate levels in the CQ-treated groups.

It is worth noting that CQ administration resulted in a three-fold decrease in the level of betaine in the blood serum. Betaine is a trimethyl derivative of glycine associated with homocysteine metabolism [40], and is normally present in plasma due to endogenous synthesis in the liver via oxidation of choline. Betaine is an effective methyl donor, and is further metabolized in vivo into dimethylglycine, sarcosine, and finally into glycine. The betaine concentration was found to have a correlation with the homocysteine concentration [41]. One of the possible mechanisms of a strong decrease in the concentration of serum betaine after CQ treatment may be its effect on homocysteine metabolism in the liver, in particular on betaine homocysteine S-methyltransferase, localized on autolysosomal membranes [42]. Our data suggest that intake of CQ by patients may be accompanied by hyperhomocysteinemia, a risk factor for vascular complications and neurodegenerative diseases [41].

Our results demonstrate that CQ administration induces metabolomic changes both in fasting rats and in animals without food deprivation. As expected, the metabolomic profiles of 48 h fasting rats treated with CQ differed from those of 48 h fasting groups without autophagy inhibition. In particular, the levels of 25 metabolites differed according to the analysis of variances between the fasting and CQ + fasting groups. However, it is worth noting that the CQ administration shifts the metabolomic profile much weaker than the fasting regimen. For example, the CQ administration does not decrease the level of butyrates and ketone bodies, with the exception of 3-hydroxybutyrate: the concentration of this metabolite decreases 2.6-fold (from 19 to 7.2 mM) in Wistar and 1.4-fold (from 21 to 15 mM) in OXYS rats. MSEA (Figure 5) demonstrates that while both fasting and CQ administration affect pathways associated with the metabolism of similar amino acids, CQ does not affect fatty acid metabolism. In summary, our results indicate that inhibition of autophagy by CQ significantly affects metabolic profiles per se, while it has little effect on the profound changes in serum metabolite level induced by food deprivation.

### 3.3. Differences in the Metabolic Response to Modulation of Autophagy between Senescence-Accelerated OXYS and Control Wistar Rats

As shown by two-way ANOVA, for many metabolites the effects of autophagy modulation depended on the animal genotype, indicating a dysregulation of metabolome reactivity in OXYS rats. Thus, the metabolic responses to modulation of autophagy in OXYS rats and Wistar rats are different. Altered metabolites in OXYS rats may serve as potential biomarkers of the manifestation of the signs of accelerated aging.

Up to the age of 4 months, OXYS rats spontaneously develop first signs of accelerated-senescence syndrome, which is characterized by early development of a phenotype similar to human geriatric disorders. This pathological phenotype primarily includes cataract, accelerated thymus involution, cognitive decline with relevance to abnormalities in Alzheimer disease, retinopathy similar to age-related macular degeneration, high blood pressure, and senile osteoporosis [27,28,29,30,31]. Metabolomic analysis performed in this work demonstrates that untreated 4-month-old OXYS rats showed significantly lower levels of glutamate, lactate, citrate, and cytidine and a higher level of sarcosine in serum as compared to Wistar rats. Moreover, two-way ANOVA showed a significant impact of the OXYS genotype on overall higher levels of branched-chain amino acids (BCAAs: valine, leucine, and isoleucine) tyrosine, histidine, and acetate and decreased concentration of pyruvate and 3-methyl-2-oxovalerate in all five experimental groups. We also observed a strain-specific differential response to both 48 h fasting and CQ exposure for the following metabolites: acetylcarnitine, serine, isoleucine, glycine, AABA, 2-ketoisovalerate, ketoleucine, glucose, and α-/β-mannose.

The increased level of BCAAs in OXYS rats may indicate the impaired autophagy: the elevated circulating levels of BCAAs have recently been reported for atg4b^−/−^ mice with autophagy deficiency [43]. The blood levels of BCAAs are also associated with insulin resistance and maple syrup disease [44]. The BCAAs are potent agonists of the amino acid sensitive mechanistic target of rapamycin complex 1 (mTORC1) protein kinase, a central regulator of metabolism and aging, and a strong negative regulator of life span in mice and many model organisms [44]. In rats, the negative effect of BCAA excess on insulin sensitivity is accompanied by increased mTORC1 activity in skeletal muscle, and this BCAA-induced insulin resistance is reversible by rapamycin, an acute inhibitor of mTORC1 [45]. Human data suggest that plasma BCAAs are associated with an increased risk of age-associated diseases [46]. In addition to BCAAs, we revealed OXYS-specific changes in the levels of branched chain α-keto acids (BCKAs), 2-ketoisovalerate and ketoleucine. The catabolism of BCAA and BCKA is linked to the functioning of branched-chain alpha-keto acid dehydrogenase complex (BCKDC) [47], and the observed effects may indicate the impairment of this process in OXYS rats. Interestingly, autophagy-induced changes in the serum metabolome profile of OXYS rats are similar to changes observed for autophagy-deficient mice. Similarly to autophagy-deficient mice [43], OXYS rats showed changes in the levels of certain amino acids involved in aminoacyl-tRNA biosynthesis and metabolytes from glycolysis and the citric acid cycle. We hypothesize that changes in metabolic pathways for BCAA metabolism, energy metabolic pathways, and aminoacyl-tRNA biosynthesis in OXYS rats indicate alternations in the autophagy, related to the formation of a complex of signs of premature aging. Further biochemical and molecular research are required in this direction.

## 4. Materials and Methods

### 4.1. Materials

All chemicals were purchased from Sigma-Aldrich (St. Louis, MI, USA). Phosphate-buffered saline (PBS) was purchased from Biolot (Moscow, Russia). H_2_O was deionized using an ultra-pure water system (SG water, Munich, Germany) to the quality of 18.2 MΩ. D_2_O 99.9% and sodium 4,4-dimethyl-4-silapentane-1-sulfonate (DSS) were purchased from Cambridge Isotope Laboratories Inc. (Tewksbury, MA, USA). 

### 4.2. Animals

Male Wistar and senescence-accelerated OXYS rats at 4 months of age were purchased from the Center for Genetic Resources of Laboratory Animals at the Institute of Cytology and Genetics, SB RAS. The OXYS strain was derived from the Wistar strain of rats at the Institute of Cytology and Genetics as described earlier [27]. The animals were kept under standard laboratory conditions (22 ± 2 °C, 60% relative humidity, and the 12 h light/12 h dark cycle) and had ad libitum access to standard rodent feed (PK-120-1, Laboratorsnab, Ltd., Moscow, Russia) and water unless stated otherwise. The study was conducted according to Directive 2010/63/EU of the European Parliament and of the Council of 22 September 2010 and was approved by the Commission on Bioethics at the ICG SB RAS (# 34 of 15 June 2016), Novosibirsk, Russia.

### 4.3. Fasting, PBS, and Chloroquine Treatment

OXYS and Wistar rats of the age of 4 months were randomly distributed into treatment and control groups. Autophagy was induced by fasting the animals for 48 h, and autophagy inhibition was performed with the drug chloroquine (CQ) diluted in 0.01 M phosphate-buffered saline (PBS) by intraperitoneal injection to animals for 4 days before euthanasia (50 mg/kg, with the last injection administered 3 h before euthanasia). Two control groups were used: intact control and vehicle PBS-injected rats, which served as a control for the CQ treatment groups. The rats were euthanized by CO_2_ asphyxiation and decapitated. Blood samples were collected by decapitation and kept at room temperature for 15 min, centrifuged at 3000 rpm for 15 min. Blood serum was collected from animals of the following five experimental groups (Figure 1): (1) control, intact rats, (2) rats fasting for 48 h, (3) PBS, vehicle control, (4) CQ, four daily intraperitoneal injections by 50 mg/kg, (5) combined effects of 48 h fasting during CQ treatment, four daily intraperitoneal injections by 50 mg/kg. Each group consisted of 6 samples for both strains of rats, except the OXYS fasting group which consisted of 5 rats. The collected serum samples were stored at −70 °C until analyzed.

### 4.4. Metabolite Extraction and NMR Spectroscopy

At the first stage of the study, the protocol for sample preparation of biological tissues of experimental animals was optimized (separation of blood serum and extraction of water-soluble metabolites). Based on previous experience [48,49,50], cold methanol-chloroform extraction was chosen as the most efficient method for extracting metabolites from blood serum. On the basis of preliminary experiments, the optimal ratio of the volumes of serum and extracting solution is the ratio: serum/methanol/chloroform = 1/1/1. In this work, 300 µL of cold methanol (−20 °C) and 300 µL of cold chloroform (−20 °C) were added to 300 µL of serum. Samples were mixed on a mini-vortex centrifuge and placed in a shaker for 30 min at 1300 rpm. The samples were kept in a freezer at −20 °C for 30 min, then centrifuged at 12,000 rpm at 4 °C for 30 min, and the supernatant was taken. The supernatant was dried on a vacuum evaporator and stored at −70 °C.

Analysis of the sample metabolomic composition was carried out by ^1^H NMR spectroscopy. The dried extracts were dissolved in 600 μL of deuterated phosphate buffer (0.01 M, pH = 7.4) containing DSS at the concentration of 6 µM as an internal standard. The samples were thoroughly mixed on a vortex, then placed on a shaker for 30 min. The sample was transferred from the tube to an NMR ampule (5 mm). The spectra were recorded using an AVANCE III HD 700 MHz NMR spectrometer (Bruker BioSpin, Ettlingen, Germany) equipped with a 16.44 T Ascend cryomagnet, 5 mm TXI 1H-13C/15N/D ZGR probehead. The spectra were acquired using a single pulse *zgpr* sequence (detection pulse was 90 degrees) with water signal suppression (saturation of the water signal with low power (20 µW) continuous RF during the delay between repetitions), 14 ppm spectral width, 13 sec relaxation delay, 6.7 s acquisition time. The spectra for each sample were recorded by a total accumulation of 128 spectra. The temperature of the sample during the recording of the spectrum was maintained at 25 °C. 

### 4.5. Identification and Quantification of Metabolites

The collected NMR spectra were manually phased and baseline corrected. Signal processing and integration were performed using MestReNova V.12 (Mestrelab Research S.L.) software. We used DSS (chemical shift 0.00 ppm) as reference for chemical shift and determination of metabolite concentration. The metabolite resonance assignments were made by comparison with data of Human Metabolome Database (http://www.hmdb.ca (accessed on 12 September 2022), [51]) and our own experience in the metabolomic profiling [52,53,54,55], or by adding reference compounds whenever needed. The concentration of metabolites in the samples was calculated by integration of the peak area of the metabolite to DSS signal; a detailed description of concentration determination is published in the work [56]. 

### 4.6. Data Analysis

Principal component analysis (PCA) and sparse partial least squares discriminant analysis (sPLS-DA) were performed using normalized metabolite concentration data to display the general metabolomic differences in the data and to analyze the contributions of all metabolites into metabolic fingerprints. The meaningful patterns of metabolite concentration changes were determined by Metabolite Set Enrichment Analysis (MSEA). PCA scores plots, Volcano plots, and MSEA plots were constructed on the MetaboAnalyst 5.0 web-platform (www.metaboanalyst.ca (accessed on 12 September 2022), [57]) with Statistical Analysis (one factor) module and Enrichment Analysis module, respectively.

### 4.7. Statistics

Statistical analysis was performed using GraphPad Prism 9.3.1 (San Diego, CA, USA). We used one-way and two-way analysis of variance (ANOVA), as well as nonparametric statistical methods. The correspondence of distributions to the normality conditions was checked by the Shapiro–Wilk test. The homogeneity of dispersions was assessed using the Levene’s test. For normally distributed samples with uniform variances, ANOVA analysis of variance was used with post hoc multiple comparisons test. Two-way ANOVA considered genotype (Wistar, OXYS) and treatment (intact control, fasting induction, PBS, chloroquine, chloroquine inhibition) as independent factors. The influence of individual factors (control vs. fasting, PBS vs. CQ, fasting vs. chloroquine inhibition) was also analyzed using one-way ANOVA. Statistical differences for samples with heterogeneous variances were analyzed using the nonparametric Kruskall–Wallis rank test with post hoc comparisons in the Mann–Whitney test. It should be noted that parametric and nonparametric methods of statistics led to similar results. The results were considered statistically significant at *p* < 0.05.

## 5. Conclusions

In this study, we compared the metabolic responses to autophagy modulation in OXYS rats with a senescent-accelerated phenotype and in control Wistar rats. Altered metabolites in OXYS rats may serve as potential biomarkers of the manifestation of accelerated aging. We also revealed metabolic signatures associated with 48 h fasting and CQ administration which could be useful subsequently when assessing the effects of taking CQ and its derivatives, as well as prolonged fasting. Our findings may contribute to a better understanding of the mechanism that links metabolism and autophagy.

## Figures and Tables

**Figure 1 ijms-23-12720-f001:**
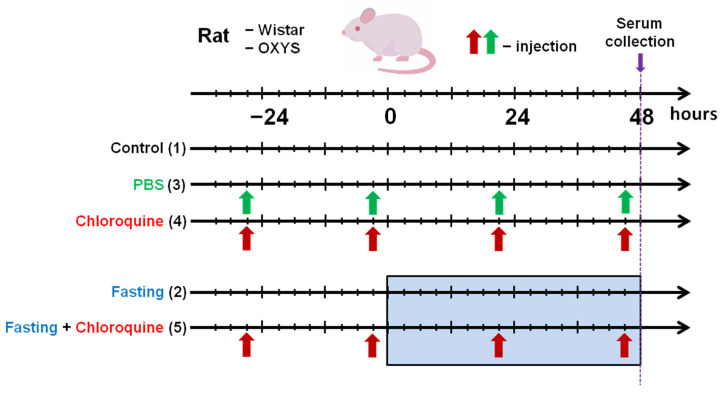
Autophagy experimental design. Five groups of each strain of rats were used: (1) control; (2) 48 h fasting; (3) PBS, i/p injected, vehicle control; (4) CQ, i/p injected; (5) CQ + fasting.

**Figure 2 ijms-23-12720-f002:**
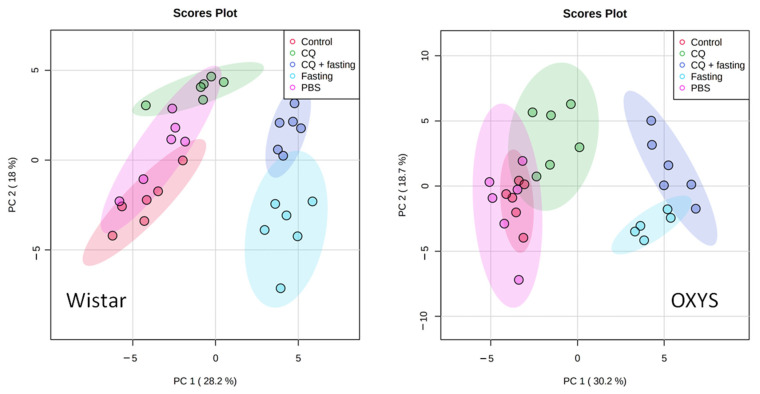
PCA score plots for concentration of aqueous metabolites extracted from blood serum of two rat strains. The groups are presented according to experimental design (Figure 1, Appendix A).

**Figure 3 ijms-23-12720-f003:**
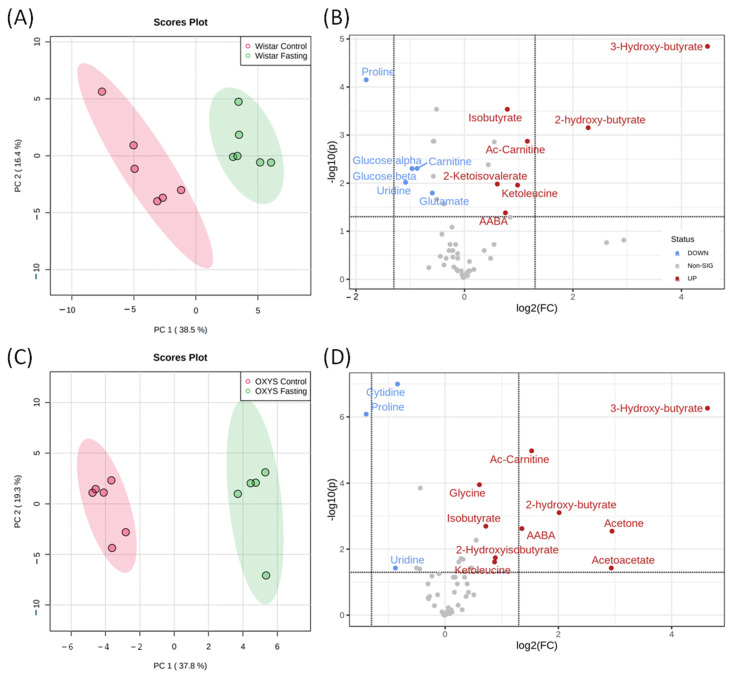
PCA score plots and Volcano plots of blood metabolites of Wistar (**A**,**B**) and OXYS (**C**,**D**) rat strains for fasting and control groups. In Volcano plots, the horizontal line depicts a cut-off of FDR-adjusted *p*-value = 0.05; metabolites with fold change threshold (fasting vs. control) = 1.5 are highlighted.

**Figure 4 ijms-23-12720-f004:**
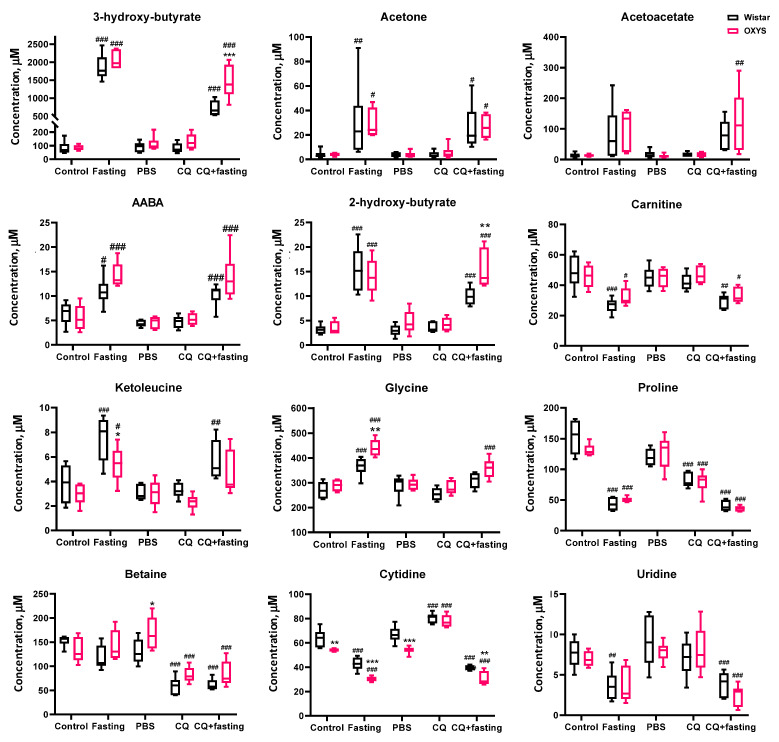
Concentrations of some serum metabolites for Wistar (black boxes) and OXYS (pink boxes) rats under autophagy modulation by 48 h fasting, chloroquine (CQ) treatment, and combined effects of fasting and CQ. The complete set of boxplots of altered serum metabolites is given in Appendix A. Data presented by boxplots: medians, the 25–75% interquartile range (bars), and min–max (error bars). Intact control serves as a control group in the case of fasting, and phosphate-buffered saline (PBS) group serves as a vehicle control to CQ and CQ + fasting treatment groups. Notation means: * *p* < 0.05, ** *p* < 0.01, *** *p* < 0.001 OXYS vs. Wistar rats from the same treatment group; # *p* < 0.05, ## *p* < 0.01, ### *p* < 0.001 vs. control group (intact control in the case of fasting, PBS vehicle control in the case of CQ/CQ + fasting) within the same strain of rats, post hoc comparisons after ANOVA or Kruskall–Wallis test.

**Figure 5 ijms-23-12720-f005:**
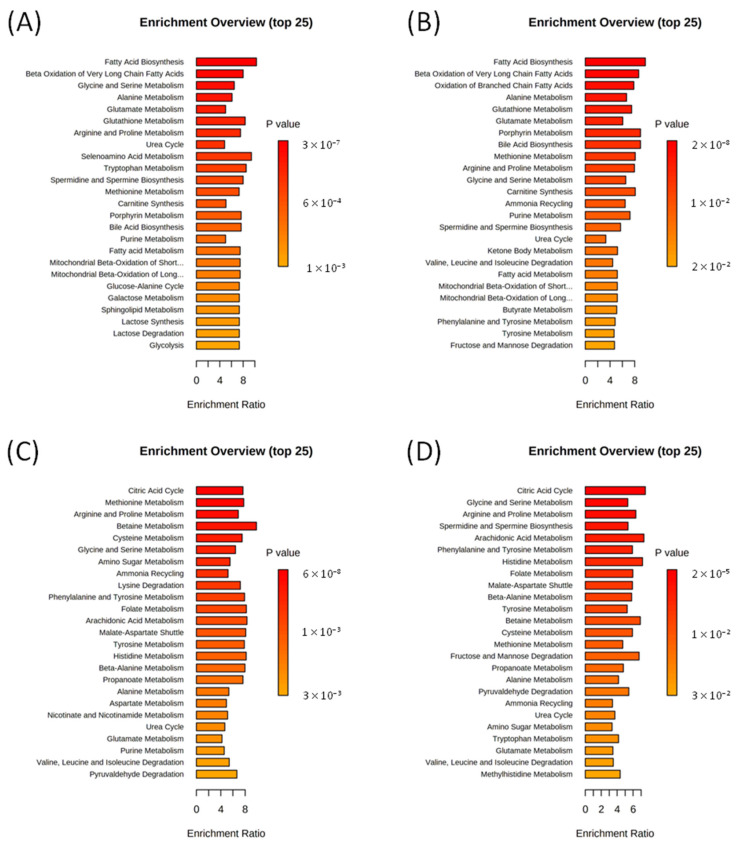
Summary plots for quantitative enrichment analysis (MSEA) based on the comparison of serum metabolomic compositions: of fasting and control Wistar (**A**) and OXYS (**B**) rats; of CQ-treated and control Wistar (**C**) and OXYS (**D**) rats.

**Figure 6 ijms-23-12720-f006:**
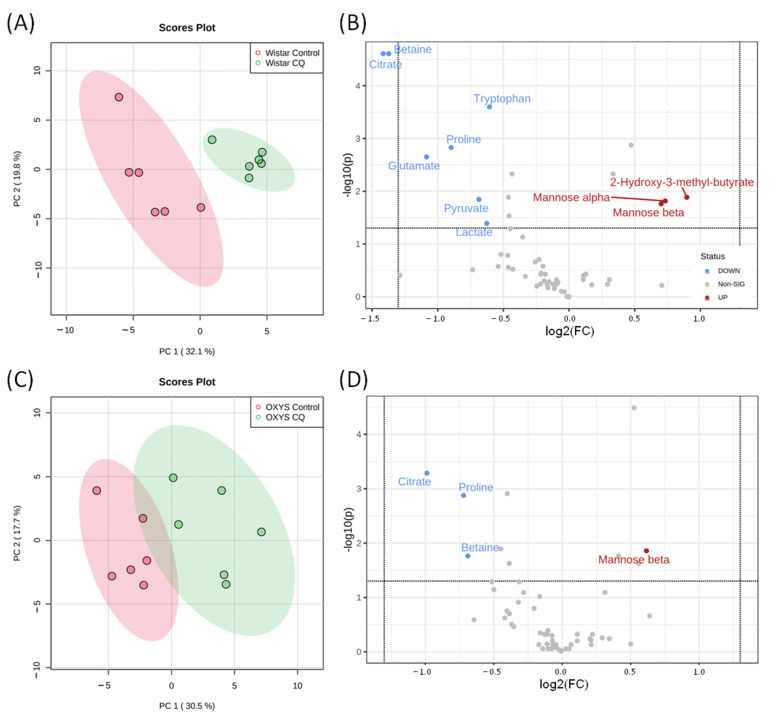
The metabolomic profiles of control and CQ treatment groups. PCA score plots and Volcano plots of blood metabolites of Wistar (**A**,**B**) and OXYS (**C**,**D**) rat strains. In Volcano plots, horizontal line depicts a cut-off of FDR-adjusted *p*-value = 0.05, metabolites with fold change threshold (CQ treatment vs. control) = 1.5 are highlighted.

**Figure 7 ijms-23-12720-f007:**
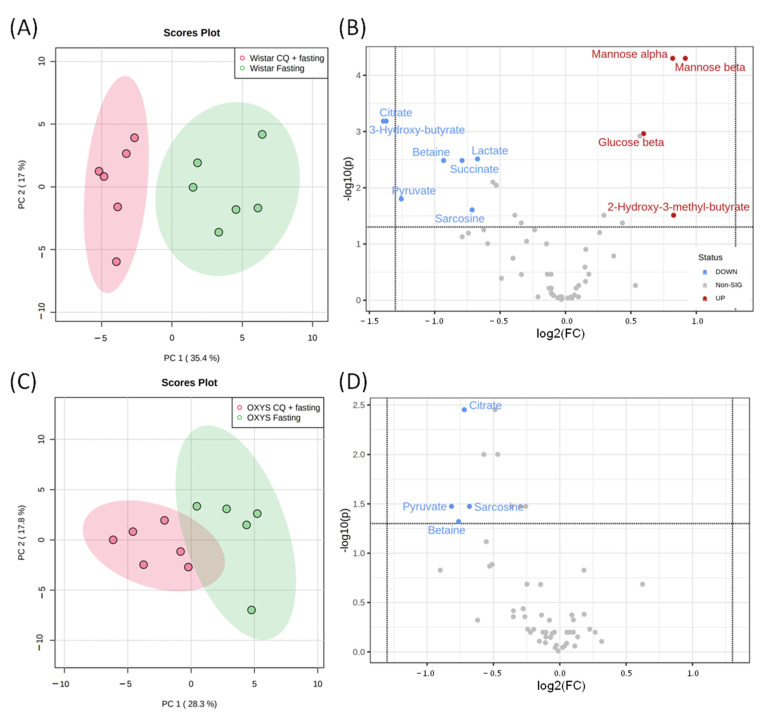
The metabolomic profiles of fasting groups with and without CQ treatment. PCA score plots and Volcano plots of blood metabolites of Wistar (**A**,**B**) and OXYS (**C**,**D**) rat strains. In Volcano plots horizontal line depicts a cut-off of FDR-adjusted *p*-value = 0.05, metabolites with fold change threshold (CQ treatment vs. fasting) = 1.5 are highlighted.

## Data Availability

Raw NMR spectra, description of specimens, and sample and metabolite concentrations are available at the Animal Metabolite Database repository, Experiment IDs 224 (https://amdb.online/amdb/experiments/224/ (accessed on 12 September 2022)). All obtained data are available from the corresponding author upon request.

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
