# Peer review of "Quantitative Metabolomic Analysis of Changes in the Rat Blood Serum during Autophagy Modulation: A Focus on Accelerated Senescence"

_ijms, 2022, doi:10.3390/ijms232112720_

Round 1

Reviewer 1 Report

In this manuscript Snytnikova et al. report a characterization of metabolite profiles of rat samples obtained for rats with different genetic background, and interventions (fasting, CQ treatment, CQ+fasting) and compared them to the metabolic profiles obtained for healthy control rats. The manuscript is interesting and could contribute to the molecular understanding of autophagy regulation.

There are a few issues that should be addressed by the authors:

·       In line 99 the authors mention that they determined absolute concentrations of the metabolites, however, in the methods section they mention normalization of NMR data. This needs to be clarified, because the quantitative information gets lost when the spectra are normalized.

·       In addition to PCA analysis the authors should carry out a PLS-DA and O-PLS-DA (with cross-validation) analysis. This can be done easily in MetaboAnalyst which the authors used to prepare the PCA and volcano plots. Analysis of significance is more obvious in O-PLS-DA analysis than in PCA analysis.

·       Figure 4: just reading the legend it is unclear to me which samples belong to which genetic background. The same is true for the significances label with stars and hashes.

·       Methods 4.4: the NMR methods need to be explained in more detail, which probehead was used, what was the spectral width, relaxation delay etc.. I was wondering if presaturation (partially) suppresses the H1’ beta glucose signal close to the water signal and should be excluded from the analysis?

·       When reading the abstract and the discussion, it was somewhat unclear to me what the motivation and the main conclusions of your work are. Please provide a clear rationale for the study and a clear conclusion. It seems that CQ reverses the fasting phenotype for some metabolites, and that there are differences between the genetic background. It would improve the readability if the authors arrange the discussion and the mechanistic speculations around the rationale and the key conclusions.

Minor points:

·       The abbreviation AABA is used in the manuscript (but not introduced; only in the list of abbreviations), please use the full name throughout.

·       Line 161: isobutirate should be isobutyrate

·       Line 161: “and so on”, please use a different term

·       Line 194: the authors mention ascorbate, but I didn’t find this compound in the figures/tables

·       Line 253: what do you mean with “ketose” ?

·       Entire paragraph from line 285-327: this paragraph needs proof reading/correction of English

Author Response

We thank you for  useful comments which help us to improve the quality of the paper. Please find below our point-to-point responses to the comments.

Comment: In line 99 the authors mention that they determined absolute concentrations of the metabolites, however, in the methods section they mention normalization of NMR data. This needs to be clarified, because the quantitative information gets lost when the spectra are normalized.

Response: We presented the absolute values of metabolite concentrations (in units of µM) in Tables SI2 and SI3. However, the data normalization was used for constructing the PCA plots. This step is needed to equalize the contributions of major and minor metabolites; otherwise, PCA scores plots would be based on concentrations of major metabolites only.

Comment: In addition to PCA analysis the authors should carry out a PLS-DA and O-PLS-DA (with cross-validation) analysis. This can be done easily in MetaboAnalyst which the authors used to prepare the PCA and volcano plots. Analysis of significance is more obvious in O-PLS-DA analysis than in PCA analysis.

Response: PLS-DA and O-PLS-DA plots are usually used for a better visualization of a difference between groups. In our case, the difference is already well visible at unsupervised PCA plots, so there is no need to use PLS-DA or O-PLS-DA. The exception is Figure 2, where the difference between groups is not pronounced. In the revised version of the manuscript, we added to SI PLS-DA analysis for the data presented in Figure 2.

Comment: Figure 4: just reading the legend it is unclear to me which samples belong to which genetic background. The same is true for the significances label with stars and hashes.

Response: The legend to Figure 4 is corrected to make it clearer.

Comment: Methods 4.4: the NMR methods need to be explained in more detail, which probehead was used, what was the spectral width, relaxation delay etc.. I was wondering if presaturation (partially) suppresses the H1’ beta glucose signal close to the water signal and should be excluded from the analysis?

Response: We added the details of the experimental setup into “Materials and Methods” section. As related to the beta glucose signal saturation: the power of the presaturation pulse was 20 µW, which corresponds to the spectral with of 30 Hz. The distance between water signal and beta glucose signal under experimental conditions was 0.143 ppm (100 Hz), so this signal is practically unaffected by saturation.

Comment: When reading the abstract and the discussion, it was somewhat unclear to me what the motivation and the main conclusions of your work are. Please provide a clear rationale for the study and a clear conclusion. It seems that CQ reverses the fasting phenotype for some metabolites, and that there are differences between the genetic background. It would improve the readability if the authors arrange the discussion and the mechanistic speculations around the rationale and the key conclusions.

Response: We added the following paragraph into discussion: “The aim of this study was to investigate the changes in metabolomic composition of blood serum during the induction and inhibition of autophagy process. We used two strains of Wistar and OXYS rats, the latter being a unique genetic model of accelerated senescence and the development of age-related diseases [33]. This allowed us to demonstrate changes in the blood serum metabolome not only under the condition of autophagy processes, but also in pathology. To our knowledge, this work is the first report on the serum metabolome of fasting rats studied by 1H NMR spectroscopy. We assessed the alterations of quantitative content of 55 serum metabolites, including amino acids, organic acids, antioxidants, osmolytes, glycosides, purine and pyrimidine derivatives. Below we discuss the obtained results of effects of autophagy modulation by fasting and CQ treatment on serum metabolome, as well as differences in the metabolic response to modulation of autophagy between senescence-accelerated OXYS rat strain and Wistar rats with normal aging rate.”

Comment: Minor points:

The abbreviation AABA is used in the manuscript (but not introduced; only in the list of abbreviations), please use the full name throughout.

Line 161: isobutirate should be isobutyrate

Line 161: “and so on”, please use a different term

Line 194: the authors mention ascorbate, but I didn’t find this compound in the figures/tables

Line 253: what do you mean with “ketose” ?

Response: Thanks for pointing these mistakes out, now they are corrected.

Comment: Entire paragraph from line 285-327: this paragraph needs proof reading/correction of English

Response: The mentioned text was corrected.

Sincerely,

Snytnikova Olga

Reviewer 2 Report

The authors have provided a valuable study in this field.

The introduction is well-written, results are properly shown and discussed.

Minor corrections in grammar and syntax are required.

References could be cited more.

Author Response

We thank you for comments  to  the paper. Please find below our  responses to the comment.

Comment: References could be cited more.

Response: We agree that autophagy is a process, which lately attracts much attention, and there are hundreds of papers recently published on that issue. However, we already have 57 references cited in the manuscript, and additional citing will step over the decent boundaries for a research article.

Sincerely,

Snytnikova Olga